# Efficacy of Nano and Conventional Zinc and Silicon Fertilizers for Nutrient Use Efficiency and Yield Benefits in Maize Under Saline Field Conditions

**DOI:** 10.3390/plants14050673

**Published:** 2025-02-22

**Authors:** Abbas Shoukat, Uswah Maryam, Britta Pitann, Muhammad Mubashar Zafar, Allah Nawaz, Waseem Hassan, Mahmoud F. Seleiman, Zulfiqar Ahmad Saqib, Karl H. Mühling

**Affiliations:** 1Institute of Plant Nutrition and Soil Science, Kiel University, Hermann-Rodewald-Str. 2, 24118 Kiel, Germany; ashoukat@plantnutrition.uni-kiel.de (A.S.); bpitann@plantnutrition.uni-kiel.de (B.P.); 2Institute of Soil and Environmental Sciences, University of Agriculture, Faisalabad 38040, Pakistan; 3Institute of Agricultural and Resource Economics, University of Agriculture, Faisalabad 38040, Pakistan; uswahmaryam676@gmail.com; 4Department of Plant Breeding and Genetics, University of Agriculture Faisalabad, Faisalabad 38040, Pakistan; m.mubasharzafar@gmail.com; 5Institute of Soil Chemistry & Environmental Sciences, AARI, Faisalabad 38040, Pakistan; an_uaf99@yahoo.com; 6Soil and Water Testing Laboratory for Research, Bahawalpur 63100, Pakistan; wasihraja@gmail.com; 7Department of Plant Production, College of Food and Agriculture Sciences, King Saud University, P.O. Box 2460, Riyadh 11451, Saudi Arabia; mseleiman@ksu.edu.sa

**Keywords:** nanofertilizers, maize, soil salinity, zinc, silicon, nutrient uptake

## Abstract

The increasing severity of salinity stress, exacerbated by climate change, poses significant challenges to sustainable agriculture, particularly in salt-affected regions. Soil salinity, impacting approximately 20% of irrigated lands, severely reduces crop productivity by disrupting plants’ physiological and biochemical processes. This study evaluates the effectiveness of zinc (Zn) and silicon (Si) nanofertilizers in improving maize (*Zea mays* L.) growth, nutrient uptake, and yield under both saline and non-saline field conditions. ZnO nanoparticles (NPs) were synthesized via the co-precipitation method due to its ability to produce highly pure and uniform particles, while the sol–gel method was chosen for SiO_2_ NPs to ensure precise control over the particle size and enhanced surface activity. The NPs were characterized using UV-Vis spectroscopy, XRD, SEM, and TEM-EDX, confirming their crystalline nature, morphology, and nanoscale size (ZnO~12 nm, SiO_2_~15 nm). A split-plot field experiment was conducted to assess the effects of the nano and conventional Zn and Si fertilizers. Zn was applied at 10 ppm (22.5 kg/ha) and Si at 90 ppm (201 kg/ha). Various agronomic, chemical, and physiological parameters were then evaluated. The results demonstrated that nano Zn/Si significantly enhanced the cob length and grain yield. Nano Si led to the highest biomass increase (110%) and improved the nutrient use efficiency by 105% under saline and 110% under non-saline conditions compared to the control. Under saline stress, nano Zn/Si improved the nutrient uptake efficiency, reduced sodium accumulation, and increased the grain yield by 66% and 106%, respectively, compared to the control. A Principal Component Analysis (PCA) highlighted a strong correlation between nano Zn/Si applications with the harvest index and Si contents in shoots, along with other physiological and yield attributes. These findings highlight that nanotechnology-based fertilizers can mitigate salinity stress and enhance crop productivity, providing a promising strategy for sustainable agriculture in salt-affected soils.

## 1. Introduction

The rapid growth of the global population, projected to reach nearly 10 billion by 2050, has placed immense pressure on agricultural systems to ensure food security [1]. The adverse effects of climate change exacerbate environmental stresses such as soil salinity, making the challenge even more severe. Salinity affects approximately 20% of cultivated land and 33% of irrigated land globally, significantly reducing crop productivity and threatening the livelihoods of millions who are dependent on agriculture [2,3]. This issue is particularly acute in arid and semi-arid regions, which encompass over 833 million hectares or 8.7% of the world’s soils [4,5]. High salinity levels are primarily caused by the accumulation of soluble salts like sodium chloride (NaCl), which disrupt plants’ physiological, morphological, and biochemical functions, resulting in osmotic and oxidative stress, hormonal imbalances, and yield reductions of 31–47% in crops such as maize, rice, and safflower [4].

Salinity reduces crop yield and disrupts nutrient uptake, leading to a lower nutrient use efficiency (NUE). Studies show that the phosphorous use efficiency in cassava decreases by 43% under salt stress [6]. In maize, severe salinity can cause yield losses ranging from 35% to 50% [7,8]. To meet the growing demand for food in the face of increasing environmental challenges, innovative solutions are essential to enhance crop resilience and NUE. However, the application of nanotechnology in agriculture has shown the potential to mitigate these impacts. Nanofertilizers have demonstrated the ability to enhance NUE by as much as 20–30% and improve crop yields by up to 25% under saline conditions [9,10]. Nanotechnology, with its advanced capabilities in nutrient delivery and stress alleviation, has emerged as a promising tool for addressing these issues [11]. By integrating nanoparticles into nutrient management strategies, agriculture can effectively combat salinity stress and improve productivity, particularly in regions where conventional methods are insufficient.

In agriculture, nanotechnology has transformed traditional practices into precision farming through tools such as nanofertilizers [12,13], nanosensors [14], nano-amendments, and nanopesticides [15]. These technologies not only enhance agricultural productivity but also mitigate environmental impacts and improve crop resilience under stress [16]. Nanomaterials, including nano-selenium, silica NPs, and nano Zn, have shown great potential as anti-stress agents by reducing oxidative stress, maintaining ROS homeostasis, and alleviating ionic and osmotic imbalances in salt-stressed crops [17,18]. The purity levels of these nanoparticles vary based on their synthesis methods and sources. For example, biological synthesis methods have achieved high-purity nano-selenium [19]. Among these, NPs of Zn and Si are particularly effective in promoting plant growth, enhancing nutrient uptake efficiency, and reducing oxidative damage in crops under abiotic stress conditions [20,21]. Nano Zn plays a critical role in enhancing antioxidant enzyme activity, reducing ROS accumulation, and improving seedling development under saline conditions [22,23], while nano Si strengthens cell walls, regulates the ion balance, and protects photosynthetic machinery in salt-stressed plants [24,25].

While nano Zn/Si fertilizers have demonstrated promising benefits in controlled environments [20,22,24], their effectiveness under real-world field conditions remains underexplored. Some field studies have examined the effects of Si and Zn NPs on the growth and yield of rice [26]. However, most research has been conducted in greenhouse or laboratory settings, which do not fully capture the complexities of real-world agricultural conditions. Given the increasing salinity stress in agricultural regions, it is crucial to investigate whether nanofertilizers can offer a practical advantage in open-field conditions.

Maize (*Zea mays* L.), one of the most important cereal crops globally, is particularly sensitive to salinity stress, which severely affects its growth, nutrient uptake, and yield [9,27]. Studies indicate that salinity stress significantly reduces the grain yield of maize by decreasing both the number of grains and their weight during the reproductive stage, ultimately leading to yield losses of up to 50% in severely affected regions [28]. However, whether nano Zn/Si fertilizers can mitigate these losses under field conditions remains uncertain.

This study aims to evaluate the impact of nano and conventional Zn and Si fertilizers on maize’s performance under saline and non-saline field conditions. We hypothesize that nano Zn/Si fertilizers may enhance the growth and yield of maize by improving its nutrient uptake and stress tolerance, particularly under salinity stress. By bridging the gap between controlled-environment research and real-field applications, this study contributes to the practical understanding of nanofertilizers’ effectiveness in sustainable agriculture.

## 2. Results

### 2.1. Agronomic Parameters

The data revealed significant effects of fertilizers under both non-saline and saline conditions on the agronomic parameters of maize (Table 1). Nano Zn showed the highest plant height (360 cm) under non-saline conditions, followed by nano Si and conven. Zn (conventional Zn), which achieved around 340 cm, while saline conditions reduced plant heights, with nano Zn/Si maintaining higher values (~300 cm) compared to the control (Figure 1a). The tassel length was highest for nano Si (45 cm) under non-saline conditions, followed by nano Zn and conven. Si (~40 cm), while under saline conditions, nano Si remained superior (Figure 1b). Similarly, nano Si achieved the maximum cob length (50 cm) under non-saline conditions, followed by nano Zn (45 cm), with reductions under saline conditions (Figure 1c). Nano Zn/Si increased the no. of cobs in both saline and non-saline conditions compared to the control (Figure 1d). The control consistently recorded the lowest values across all parameters.

### 2.2. Chemical Parameters

Plants counteract salinity stress by reducing Na⁺ accumulation through ion exclusion, compartmentalization, and enhanced antioxidant defense. In this study, the Na⁺ concentration was highest in the shoots at 110 mg/kg DW under saline conditions, while nano Zn and nano Si significantly reduced the Na⁺ accumulation, indicating their role in mitigating salt stress (Figure 2a, Table 2). For K⁺, the highest accumulation was in the nano Si-treated plants under non-saline conditions at 130 mg/kg DW, followed by nano Zn and conven. Zn. Salinity reduced the K⁺ levels, but nano Si/Zn maintained higher concentrations than the control (Figure 2b). Regarding Zn in shoots, nano Zn had the highest accumulation under non-saline conditions at 19 mg/kg, followed by nano Si at 14 mg/kg. Under salinity, the Zn levels decreased, with nano Zn still having the highest levels at 12 mg/kg (Figure 2c). The Si in shoots was greatest under nano Si, reaching 8 mg/g in non-saline and 6.5 mg/g in saline conditions (Figure 2d). In grains, the Zn was highest under nano Zn, with 34 mg/kg in non-saline and 22 mg/kg in saline conditions. Conven. Zn and nano Si also contributed to Zn accumulation (Figure 2e). Regarding Si, nano Si had the highest levels, with 62 mg/kg in non-saline and 32 mg/kg in saline conditions, followed by conven. Si (Figure 2f).

### 2.3. Nutrient Use Efficiency Parameters

Regarding Zn use efficiency, nano Zn showed a 72% increase under non-saline and a 65% increase under saline conditions compared to the control. Zn and Si were selected for this study due to their well-established roles in plant stress mitigation. Zn enhances enzymatic activity and antioxidant defense, while Si improves the cell wall integrity and reduces ionic toxicity. Conven. Zn resulted in a 22% increase under non-saline and 8.4% increase under saline conditions. Nano Si led to a 110% increase under non-saline and a 105% increase under saline conditions, while conventional Si showed an 80% increase under non-saline and a 45% increase under saline conditions (Figure 3a). Regarding Si use efficiency, nano Si exhibited a 388% increase under non-saline and a 253% increase under saline conditions, while conven. Si showed a 144% increase under non-saline and a 50% increase under saline conditions (Figure 3b). The harvest index for nano Zn showed a 41% increase under non-saline and 29% under saline conditions. Nano Si exhibited the highest increase in harvest index, with 77% under non-saline and 66% under saline conditions (Figure 3c). Regarding the partial factor productivity, nano Zn demonstrated a 63% increase under non-saline and 57% under saline conditions, while nano Si showed a 42% increase under non-saline and 40% under saline conditions (Figure 3d).

### 2.4. Yield Parameters

The results showed significant yield improvements across all treatments compared to the control under both saline and non-saline conditions (Table 1). Nano Si exhibited the highest enhancement in grain yield, with a 110% increase under non-saline and 106% under saline conditions, while nano Zn showed increases of 72% and 66% compared to the control, respectively (Figure 4a). The straw yield also showed similar improvements (Figure 4b). The biological yield increased with nano Zn and nano Si applications, surpassing the control by 22% and 19% under non-saline conditions, respectively, and by 28% and 24% under saline conditions (Figure 4c). In terms of the 100-grain weight, nano Si achieved the most significant increase, with 47% under non-saline and 120% under saline conditions compared to control. Nano Zn also showed a notable increase of 19% under non-saline and 65% under saline conditions, while conventional Zn and Si yielded intermediate results (Figure 4d).

### 2.5. Principal Component Analysis (PCA)

The PCA biplot provides insights into the relationships between different treatments and different parameters under saline and non-saline conditions (Figure 5). The first principal component (Dim 1) explains 69.3% of the total variance, while the second component (Dim 2) accounts for 17%, indicating that most of the variation in the dataset is captured along these two dimensions. The PCA plot shows a clear distinction between treatments, with nano Zn and nano Si, particularly under non-saline conditions, being positioned far from the control and conventional treatments. This suggests that these treatments had a stronger influence on the nutrient use efficiency (NUE), grain yield (GY), shoot yield (SY), and partial factor productivity (PFP). Among all treatments, nano Zn under non-saline conditions (Con+Nano Zn) and nano Si under non-saline conditions (Con+Nano Si) performed the best, as they are positioned the farthest along Dim 1, indicating a significant positive impact on plant growth and yield parameters. In contrast, the control (Con) and saline stress (Sal) treatments are clustered near the origin, reflecting their lower effectiveness in enhancing plant performance.

### 2.6. Heat Map Analysis

The heat map further illustrates the treatment-wise variations in multiple traits. The clustering pattern reveals that nano Zn/Si under non-saline conditions exhibited the highest positive influence, as indicated by the red and orange hues, which represent higher values for key parameters such as Zn uptake, cob length, and number of leaves (N leaves). This trend can be attributed to the role of Zn in enzymatic activation and protein synthesis, as well as Si’s contribution to cell wall reinforcement and stress mitigation, which together enhance growth and nutrient assimilation. Conversely, the control and conventional treatments under saline conditions show predominantly blue shades, indicating lower performances in these parameters due to the adverse effects of salinity, such as ion toxicity and osmotic stress. Nano Zn (Con+Nano Zn) and nano Si (Con+Nano Si) in non-saline conditions demonstrated the most favorable responses, while the saline-stressed control (Sal) and conventional treatments (Sal+Conv Zn, Sal+Conv Si) exhibited reduced performance, with lower nutrient uptake and biomass accumulation.

## 3. Discussion

Salinity stress is a major constraint on maize’s productivity, reducing its growth and yield by disrupting the water uptake, nutrient absorption, and metabolic stability [29]. Excess Na⁺ and Cl⁻ accumulation lowers the water potential, disrupts K⁺ homeostasis, and impairs essential functions such as photosynthesis and enzymatic activity, leading to stunted growth and reduced grain yield [2,3,29,30,31,32]. Addressing these challenges requires innovative solutions beyond conventional soil amendments and fertilizers [33,34].

This study hypothesized that nano Zn/Si fertilizers may enhance maize’s resilience to salinity stress by improving the nutrient efficiency, ionic balance, and physiological stability. Nanofertilizers have been recognized for their superior nutrient delivery, higher absorption efficiency, and stress-alleviating properties [35]. Nano Zn enhances antioxidant defense, reducing ROS accumulation and mitigating oxidative stress, while nano Si strengthens cell walls, preventing ion leakage and regulating the ionic balance by decreasing the Na⁺ uptake and increasing K⁺ absorption [29,36,37]. These mechanisms are critical for osmotic regulation and plant metabolism stability under salinity stress.

Our findings demonstrated that both nano and conventional Zn and Si fertilizers significantly influenced the growth and yield of maize under saline and non-saline conditions (Table 2 and Table 3). The superior performance of nano Zn, particularly under non-saline conditions, can be attributed to its role in activating key enzymes and enhancing auxin hormone production, which promotes cell elongation, better nutrient absorption, and overall vegetative growth [36,38,39]. This resulted in increased plant height and tassel length, leading to improved pollination efficiency and cob development (Figure 1, Table 2). Nano Si facilitated growth by improving the nutrient transport and root architecture [40]. Upon uptake through aquaporin channels, nano Si increased the root surface area, enhancing the water and nutrient absorption efficiency [41]. This improvement supported greater biomass production and reproductive development, even under saline conditions [Figure 1 and Figure 3]. Additionally, enhanced root development ensured stable plant heights and tassel growth, positively correlating with improved pollination and cob formation (Figure 1). However, an excessive number of cobs can compete for nutrients, potentially limiting the cob length and grain yield. These findings align with previous research demonstrating the efficacy of Zn fertilizers in enhancing plant height under saline conditions in maize [36,39].

Nano Zn/Si significantly reduced the Na⁺ accumulation in shoots (Figure 2, Table 2) by enhancing the membrane stability and activating H⁺-ATPase, which facilitates Na⁺ exclusion through salt-overly-sensitive 1 (SOS1) antiporters [42]. This mechanism plays a crucial role in maintaining the ionic balance under salinity stress by transporting Na⁺ out of root cells, thereby reducing its translocation to the aerial parts of the plant [42]. Si reinforced the Casparian strip, limiting Na⁺ entry, while Zn^2+^ stabilized the ion channels, improving the selective uptake [43]. The higher K⁺ levels under nano Si (Figure 2, Table 2) suggest activation of high-affinity K⁺ transporters (e.g., HAK), maintaining the K⁺/Na⁺ balance, which is crucial for enzyme function under stress [44,45]. The Zn uptake remained higher under nano Zn supply (Figure 2) due to enhanced ZIP transporter activity, ensuring better solubility and mobility despite ionic competition [46]. Si accumulation in shoots and grains (Figure 2) indicates improved stress resistance by reinforcing cell walls and reducing oxidative stress. The increased Zn and Si in grains highlights efficient translocation, improving the biofortification and stress tolerance, as shown in Figure 2 [44]. A prior study on maize’s antioxidant enzyme responses under saline conditions found that nano Zn/Si significantly enhanced enzymatic activity, reducing oxidative stress markers like MDA and H_2_O_2_ [40].

Nano Zn/Si also improved the NUE by enhancing the plants’ metabolic activity and ion regulation under stress (Figure 3) [35,46]. Nano Zn significantly increased the Zn use efficiency in saline conditions by facilitating Zn uptake through rhizospheric pH modulation, which enhanced the Zn solubility and reduced its precipitation with carbonates [47]. In saline soils, a high pH often limits Zn’s availability by promoting the formation of insoluble Zn compounds. Nano Zn application helps lower the rhizospheric pH by stimulating root exudation of organic acids and activating H⁺-ATPase, thereby maintaining Zn in a bioavailable form for uptake [47]. Nano Zn also promoted the expression of metallothioneins, reducing the Zn toxicity while maintaining optimal levels for enzymatic functions [48]. Conventional Zn, being bulkier and less soluble, exhibited a lower efficiency, as shown in Figure 3, due to limited mobility in the soil [37,38]. Nano Si exhibited the highest Si use efficiency under both saline (253%) and non-saline (388%) conditions due to its role in forming silica aggregates in cell walls, which enhanced the mechanical strength and reduced transpiration loss (Figure 3) [49]. Under saline stress, nano Si also increased the expression of high-affinity K⁺ transporters (e.g., HAK/KUP), improving the K⁺ uptake and reducing Na⁺ interference, thereby maintaining ionic homeostasis [50]. Nano Si also enhanced root exudation of organic acids, promoting Si solubilization and uptake via Lsi1 and Lsi2 transporters [51]. The harvest index was significantly higher with nano Si due to improved phloem loading of the assimilates [52], ensuring better partitioning of carbohydrates toward reproductive structures (Figure 3). The gains in partial factor productivity in nanofertilized plants were attributed to increased chlorophyll stability [53] and reduced oxidative damage, leading to prolonged photosynthetic efficiency (Figure 3).

The significant yield improvements with nano Zn/Si treatment resulted from enhanced nutrient delivery, stress mitigation, and metabolic activation (Figure 4, Table 3). Nano Si strengthened cell walls, reduced oxidative stress, and improved the water use efficiency (Table 3, Figure 4), promoting better grain filling and weight [35,37,51]. Its role in Si transport further facilitated nutrient translocation improving overall uptake efficiency. Nano Zn boosted the enzymatic activity and auxin biosynthesis, leading to increased biomass. It also regulated Na⁺-H⁺ antiporters, reducing sodium toxicity and maintaining the ionic balance under salinity stress [49,51]. The overall rise in straw and biological yield reflects improved photosynthetic efficiency and resource allocation.

The PCA revealed nano Zn/Si to be key drivers of improved NUE, yield, and biomass, explaining 69.3% and 17% of the variance, respectively. Their distinct positioning highlights their superior performance, while the control and saline treatments showed a minimal impact (Figure 5). The heat map confirmed these trends, with the nanotreatments significantly boosting the nutrient uptake and yield traits, whereas the conventional treatments under saline conditions exhibited poor performance (Figure 6). Furthermore, the morphological differences in maize cobs, reinforce these findings, demonstrating the superior effectiveness of nano Zn/Si in enhancing grain development and yield under both saline and non-saline conditions compared to conventional treatments (Figure 7). These results reinforce the potential of nano Zn/Si as effective solutions for mitigating salinity stress in maize production. This study is based on data from one year, and a second field trial is currently underway to validate the findings across different environmental conditions.

## 4. Materials and Methods

### 4.1. Nanoparticle Synthesis and Characterization

ZnO and SiO_2_ nanoparticles (NPs) were synthesized via the co-precipitation and sol–gel methods, respectively, following previously established protocols [54]. ZnO NPs were obtained by reacting NaOH with ZnSO_4_·7H_2_O at 80 °C, while SiO_2_ NPs were synthesized using TEOS and ethanol at room temperature. The characterization, conducted at Kiel University, Germany, involved UV-Vis Spectroscopy, XRD, SEM, and TEM-EDX. The analyses confirmed the crystalline nature and primarily spherical morphology of the NPs, with ZnO averaging 12 nm and SiO_2_ around 15 nm, exhibiting some agglomeration [54].

### 4.2. Study Site and Pre-Experiment Soil Analysis

The saline field, located at the PARS Campus, UAF, was classified as fine loamy, mixed, superactive, thermic Typic Haplustept (sandy clay loam soil). The normal field, located at the Soil Science Field, Main Campus, UAF, was classified as coarse loamy, mixed, superactive, thermic Typic Ustifluvent (sandy loam soil) according to the USDA Soil Taxonomy. The GPS coordinates for the normal field were 31.43473° N, 73.06858° E, while the saline field was located at 31.43510° N, 73.07025° E. Prior to sowing, surface soil samples were collected from a depth of 0–15 cm, air-dried, and sieved through a 2 mm mesh. The soil pH was determined using a Beckman pH meter in a saturated soil paste, while the electrical conductivity (EC) was measured using a digital EC meter in the saturation extract. The soil texture was determined using the hydrometer method [55]. DTPA-extractable Zn was quantified at 0.72 mg kg^−1^ using an atomic absorption spectrophotometer [56]. The detailed soil properties are provided in Table 3.

### 4.3. Soil Preparation and Fertilizer Application

The field was irrigated with canal water and plowed using tractor-drawn implements. Fertilizers were applied at recommended rates based on prior studies and optimized through preliminary pot experiments: urea (175 kg N/ha), diammonium phosphate (90 kg P/ha), and sulfate of potash (125 kg K/ha). Nano Zn and nano Si were applied at rates of 10 ppm (22.5 kg/ha) and 90 ppm (201 kg/ha), respectively, following dose selection from prior research. Conventional Zn and Si were applied at equivalent rates. Fertilizer applications were made after maize plants reached the 4-leaf stage.

### 4.4. Planting and Crop Management

Maize seeds were sown on ridges with a row-to-row distance of 75 cm and a plant-to-plant spacing of 30 cm. Twenty days after sowing, thinning was performed to ensure a uniform plant density. Broadleaf weeds were controlled at the V4 stage using Gengwei (750–1000 mL/acre) a pre-emergence herbicide containing Mesotrione and Atrazine, manufactured by Syngenta, Basel, Switzerland. Lufenuron was applied to control shoot fly borers before tasseling. Sucking pests (whitefly and aphids) were managed using the “Polytrin C” and Virtako insecticides, both manufactured by Syngenta, Basel, Switzerland.

### 4.5. Harvesting and Agronomic Measurements

At crop maturity, plants were manually harvested. Agronomic parameters including plant height, stem diameter, cob length, and cob diameter were measured. The total fresh biomass was recorded, and plant parts (leaves, cobs, and stover) were air-dried and oven-dried to a constant weight. Grains were manually separated, and their dry weight was recorded. Biomass and grain yield were calculated on a per-hectare basis using standard formulas.

### 4.6. Chemical Analysis of Plant Samples

Plant samples, including grains, were oven-dried, ground, and digested for ionic analysis. The Zn concentrations in plant tissues and grains were measured using atomic absorption spectrophotometry following wet digestion with a diacid mixture (HNO_3_:HClO_4_, 3:1). The Si contents in both plant tissues and grains were determined using the molybdenum blue method [57]. Sodium (Na⁺) and potassium (K⁺) were quantified using a flame photometer.

### 4.7. Nutrient Uptake and Efficiency

The nutrient use efficiency (NUE) was determined to evaluate the effectiveness of nutrient utilization:$$NUE=\frac{G_{f}-G_{u}}{N_{a}}$$
where *NUE* = nutrient use efficiency, *G_f_* = grain yield in fertilized plots (kg/ha), *G_u_* = grain yield in unfertilized plots (kg/ha), and *N_a_* = the amount of nutrient applied (kg/ha).

### 4.8. Yield and Quality Parameters

The biological yield was calculated as the total above-ground biomass per hectare. The grain yield was determined by threshing, cleaning, and weighing dry kernels from harvested plants in a representative area [58]. The yield per hectare was computed using the following:$$GY\left(\frac{\mathrm{kg}}{\mathrm{ha}}\right)=\frac{W_{g}}{A_{h}}\times 10,000$$
where *GY* = grain yield (kg/ha), *W_g_* = weight of grain (kg), and *A_h_* = harvest area (ha).

The 100-grain weight was calculated by randomly selecting and weighing 100 grains. The straw yield was measured by drying and weighing straw from the harvested area [58]. The harvest index (HI) was calculated using the following:
$$HI=\frac{GY}{BY}$$
where *HI* = harvest index, *GY* = grain yield (kg/ha), and *BY* = total above-ground biomass (kg/ha).

The partial factor productivity (PFP) was evaluated as the crop yield per unit of fertilizer applied [58]:$$PFP=\frac{GY}{F_{a}}$$
where *PFP* = partial factor productivity, *GY* = grain yield (kg/ha), and *F_a_* = the amount of fertilizer applied (kg/ha).

### 4.9. Experimental Design

The experiment followed a split-plot design with three replicates and five treatments across two fields [59]. Data were analyzed using Statistics 10.1 software, while R version 4.0.5 was used for advanced statistical analyses, including PCA and heatmap visualizations. Statistical comparisons were performed using ANOVA, followed by the Least Significant Difference (LSD) test to compare treatment means. The error bars in the figures represent the standard error (SE). The letters above the bars indicate significant differences among treatments based on the LSD test. In cases where multiple factors (e.g., treatment × salinity) were involved, the lettering represents the combined effect of both factors.

**Table 3 plants-14-00673-t003:** Attributes of soil used in field experiments.

Sr. No.	Name	Unit	Pars Soil	UAF Soil	Reference
1	EC_e_	dS m^−1^	9.1	1.8	[60]
2	pH_s_	----	8.1	8.4	[60]
3	CO_3_^2−^	me L^−1^	nil	nil	[61]
4	HCO_3_^−^	me L^−1^	4.1	3.8	[61]
5	Ca^2+^+Mg^2+^	me L^−1^	16.8	7.5	[62]
6	(SAR)	me L^−1^	16.89	2.1	[63]
7	Saturation percentage	Percentage (%)	37	35	[64]
8	Texture		Sandy clay loam	Sandy loam	[65]
9	Organic matter	Percentage (%)	0.83	0.9	[66]
10	Nitrogen (N)	Percentage (%)	0.061	0.05	[67]
11	Phosphorus (P)	ppm	9.32	7.8	[68]
12	Potassium (K)	ppm	129	125	[69]

## 5. Conclusions

This study shows that nano Zn and Si enhance maize’s resilience to salinity stress, improving its agronomic, chemical, and physiological parameters. Nano Si provided the highest yield increase (110% under non-saline and 106% under saline conditions) [Figure 1 and Figure 4]. Chemically, these treatments improved the nutrient uptake by reducing Na⁺ accumulation in shoots while increasing the K⁺ content, thereby optimizing the K/Na ratio, which is critical for the ionic balance and stress tolerance [Figure 2]. Nanofertilizers, especially Zn and Si, significantly enhance the nutrient use efficiency and mitigate salinity stress, as confirmed by the PCA and heat map analysis. These findings highlight the promising role of nanotechnology in sustainable agriculture, although further research is needed to evaluate long-term soil interactions, including potential effects on soil microbiology, optimize application methods, and explore the molecular mechanisms underlying stress alleviation.

## Figures and Tables

**Figure 1 plants-14-00673-f001:**
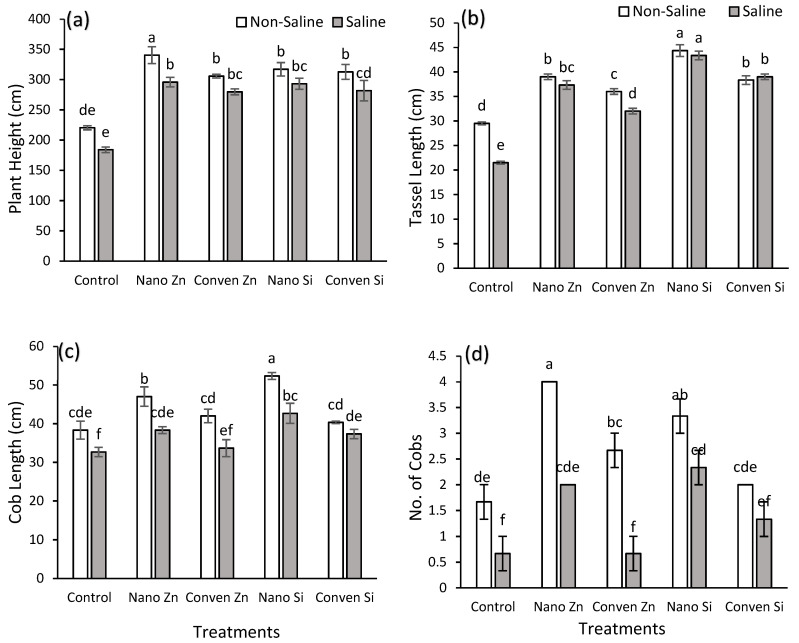
Effect of different sources of silicon and zinc on agronomic parameters under non-saline and saline conditions. (**a**) Plant height, (**b**) Tassel length, (**c**) Cob length, and (**d**) Number of cobs. Error bars represent standard error (SE), and different letters above bars indicate significant differences among treatments based on LSD test after ANOVA (α = 0.05).

**Figure 2 plants-14-00673-f002:**
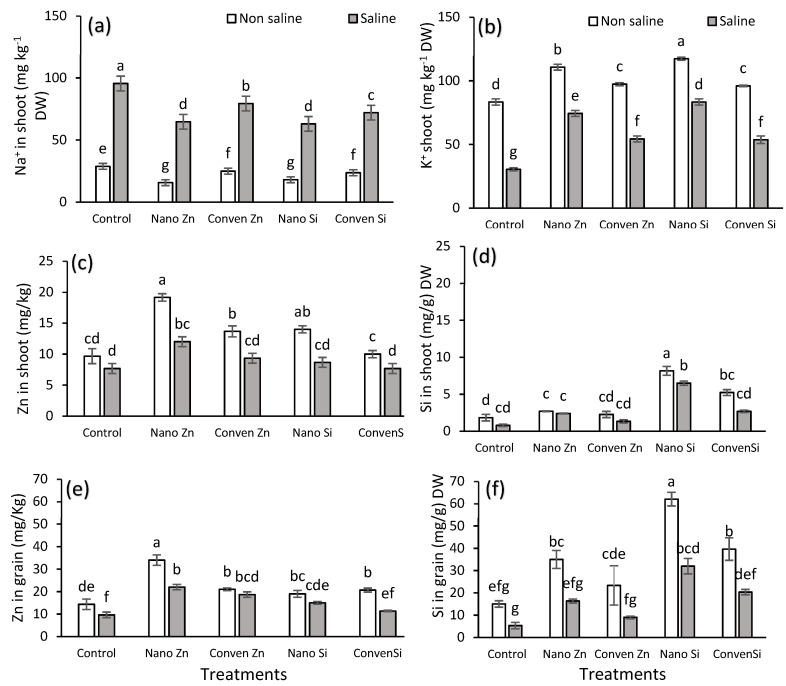
Effect of different sources of silicon and zinc on chemical parameters under non-saline and saline conditions. (**a**) Na^+^ in shoot (mg kg^−1^ DW), (**b**) K^+^ shoot (mg kg^−1^ DW), (**c**) Zn in shoot (mg/kg), (**d**) Si in shoot (mg/kg) DW, (**e**) Zn in grain (mg/kg) and (**f**) Si in grain (mg/g) DW. Error bars represent standard error (SE), and different letters above bars indicate significant differences among treatments based on LSD test after ANOVA (α = 0.05).

**Figure 3 plants-14-00673-f003:**
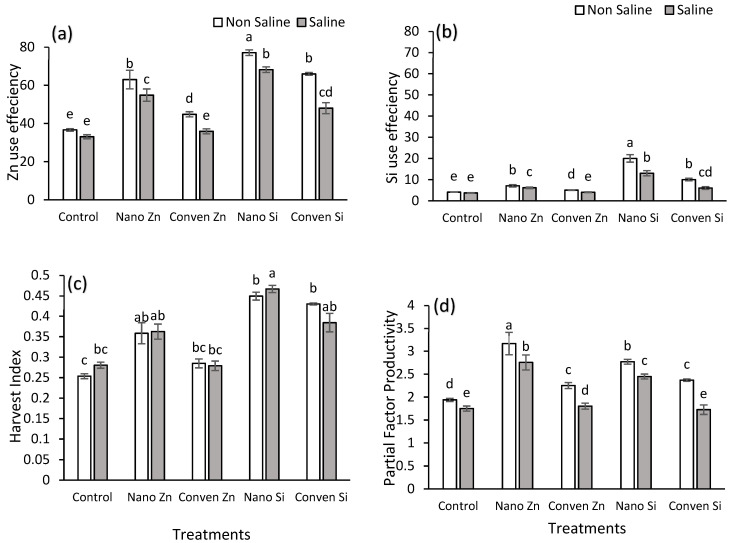
Effect of different sources of silicon and zinc on nutrient use efficiency and productivity parameters under non-saline and saline conditions. (**a**) Zn use efficiency (**b**) Si use efficiency (**c**) Harvest Index and (**d**) Partial factor productivity. Error bars represent standard error (SE), and different letters above bars indicate significant differences among treatments based on LSD test after ANOVA (α = 0.05).

**Figure 4 plants-14-00673-f004:**
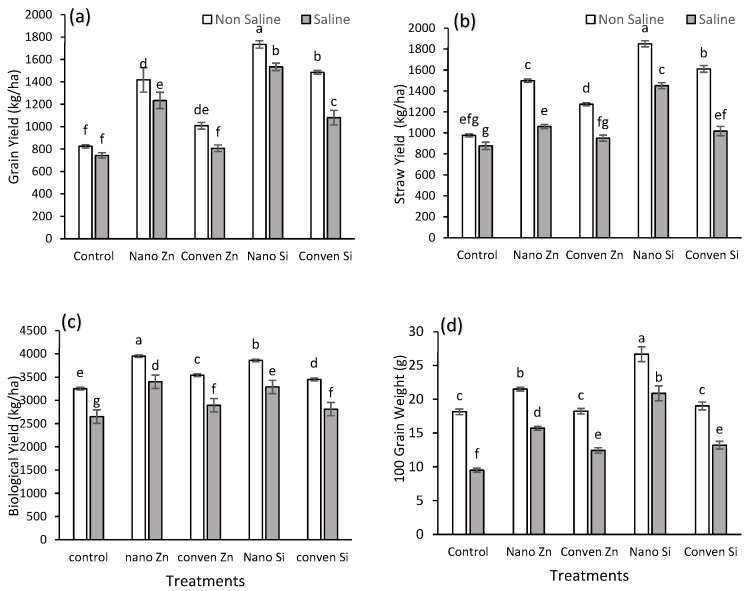
Effect of different sources of silicon and zinc on yield parameters under non-saline and saline conditions. (**a**) Grain Yield (kg/ha), (**b**) Straw Yield (kg/ha), (**c**) Biological Yield (kg/ha), and (**d**) 100 grain Weight (g). Error bars represent standard error (SE), and different letters above bars indicate significant differences among treatments based on LSD test after ANOVA (α = 0.05).

**Figure 5 plants-14-00673-f005:**
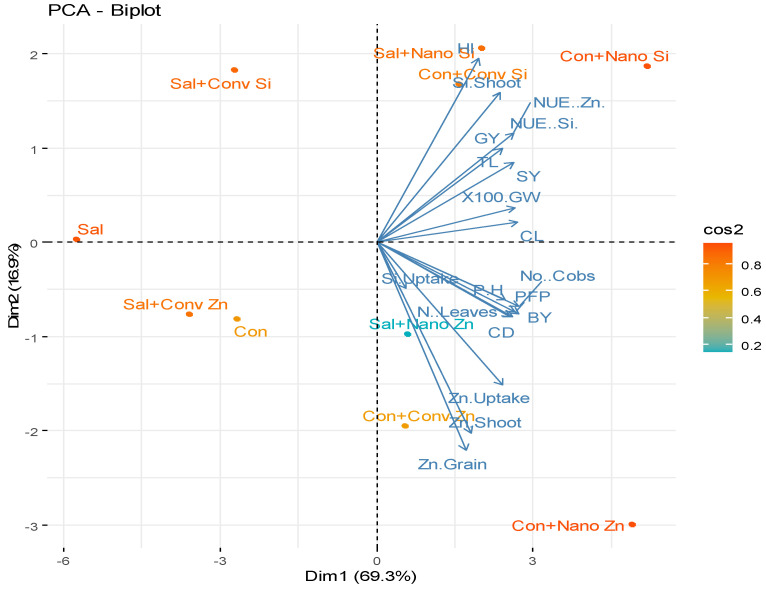
The effect of different zinc and silicon treatments on agronomic traits under non-saline and saline conditions based on Principal Component Analysis (PCA). The color scale (cos2) indicates variable representation in PCA space, with higher values (red/orange) showing better representation and lower values (blue/green) indicating weaker contribution. The parameters include salinity stress (Sal), conventional (Conv), control (Con), nutrient use efficiency (NUE), grain yield (GY), tassel length (TL), shoot yield (SY), 100-grain weight (100 GW), cob length (CL), plant height (PH), partial factor productivity (PFP), cob diameter (CD), number of leaves (N Leaves), and biological yield (BY).

**Figure 6 plants-14-00673-f006:**
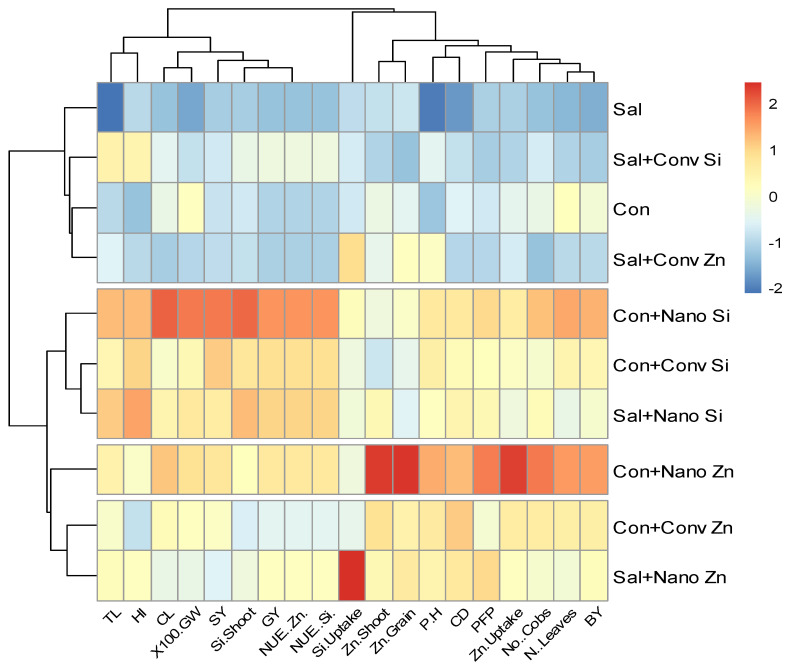
Heat map of the effect of different zinc and silicon treatments on agronomic traits under non-saline and saline conditions. The parameters include salinity stress (Sal), conventional (Conv), control (Con), nutrient use efficiency (NUE), grain yield (GY), tassel length (TL), shoot yield (SY), 100-grain weight (100 GW), cob length (CL), plant height (PH), partial factor productivity (PFP), cob diameter (CD), number of leaves (N Leaves), and biological yield (BY).

**Figure 7 plants-14-00673-f007:**
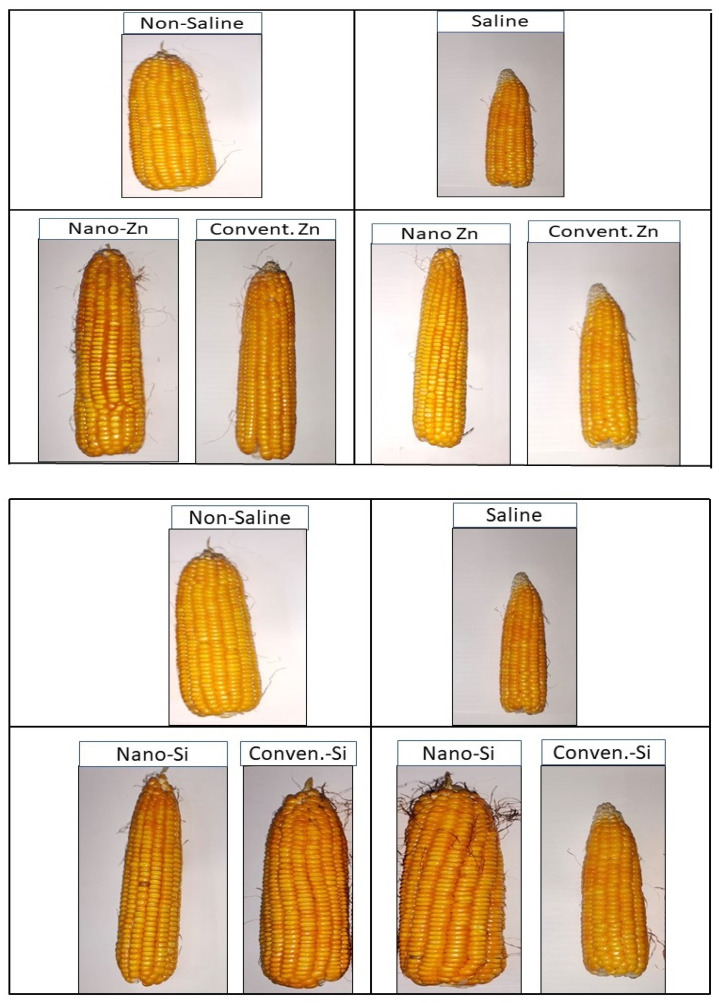
Visual representation of maize cobs subjected to different salinity levels (non-saline and saline) and nano and conventional Zn and Si treatments.

**Table 1 plants-14-00673-t001:** Effect of salinity and fertilizer treatments on agronomic and yield traits of maize (ANOVA at 0.05 Significance Level). The symbols *, **, and *** indicate significance levels at *p* < 0.05, *p* < 0.01, and *p* < 0.001, respectively. NS denotes non-significant differences. Means followed by different letters (A, B) indicate significant differences due to salinity stress (main plot effect).

Salinity	Treatments	Plant Height	Cob Length	Tassel Length	No. of Cobs	Bio. Yield	SY	GY	100 GY
**Control**	T1	220.4	38.3	29.5	1.6	3250	976	824	18.13
	T2	340.3	47	39	4	3950	1497	1416	21.5
	T3	305.7	42	36	2.6	3540	1273	1007	18.22
	T4	307	52.3	44.3	3.3	3860	1850	1733	26.66
	T5	301.6	40.3	38.3	2	3450	1610	1483	19
	**Mean**	**294A**	**44A**	**37A**	**2.73A**	**3610A**	**1441A**	**1293A**	**20.7A**
**10 dSm^−1^**	T1	184	32.66	21.5	0.66	2650	876	743	9.5
	T2	295.6	38.33	37.33	2	3400	1060	1233	15.7
	T3	279.66	33.6	32	0.66	2893	950	806	12.42
	T4	283	42.6	43.33	2.33	3286	1450	1533	20.8
	T5	253.3	37.33	39	1.33	2810	1016	1080	13.2
	**Mean**	**259B**	**36B**	**34B**	**1.4B**	**3008B**	**1070B**	**1079B**	**14.3B**
**Significance ANOVA**									
**Salinity Stress**		*	**	*	***	***	***	***	***
**Fertilizer Treatments**		***	***	***	***	***	***	***	***
**Interaction**		NS	NS	***	NS	NS	***	NS	NS

**Table 2 plants-14-00673-t002:** Effect of salinity and fertilizer treatments on chemical and efficiency traits of maize (ANOVA at 0.05 Significance Level). The symbols *, **, and *** indicate significance levels at *p* < 0.05, *p* < 0.01, and *p* < 0.001, respectively. NS denotes non-significant differences. Means followed by different letters (A, B) indicate significant differences due to salinity stress (main plot effect).

Salinity	Treatments	Zn in Shoot	Si in Shoot	Zn in Grain	Si in Grain	Na in Shoot	K in Shoot	ZnUE	SiUE
**Control**	T1	9.6	1.8	14.3	15.3	28.8	49.6	36.6	4.0
	T2	19.15	2.7	34	35	15.6	85	62.9	7.04
	T3	13.6	2.2	21	23.3	24.9	64.2	44.7	5.01
	T4	14	8.1	19	62	18	99.3	77.0	20.6
	T5	10	5.2	20.6	39.6	23.6	57.3	65.9	13
	**Mean**	**13.3A**	**4.04A**	**21.8A**	**35A**	**22A**	**71A**	**57A**	**10A**
**10 dSm^−1^**	T1	6.3	0.8	9.6	5.33	95.5	31	33.0	3.6
	T2	12	2.4	22	16.33	64.6	58.3	54.8	6.13
	T3	9.3	1.33	18.6	9	79.3	43	35.8	4.01
	T4	8.6	6.5	15	32	63	65.3	68.14	9.66
	T5	7.6	2.68	11.3	20.33	72	50.6	48	5.9
	**Mean**	**8.8A**	**2.74B**	**15.33B**	**16.6B**	**79B**	**50B**	**47B**	**5.8B**
**Significance ANOVA**									
**Salinity Stress**		NS	*	**	***	**	**	**	**
**Fertilizer Treatments**		***	***	***	***	***	***	***	***
**Interaction**		NS	**	*	NS	NS	NS	NS	NS

## Data Availability

Data will be made available on request to the corresponding author.

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
