# Peer review of "Efficacy of Nano and Conventional Zinc and Silicon Fertilizers for Nutrient Use Efficiency and Yield Benefits in Maize Under Saline Field Conditions"

_plants, 2025, doi:10.3390/plants14050673_

Round 1

Reviewer 1 Report

Comments and Suggestions for Authors

Dear All,

This study investigates the effectiveness of nano and conventional zinc (Zn) and silicon (Si) fertilizers in improving maize growth, nutrient uptake, and yield under saline and non-saline field conditions. Soil salinity is a major concern affecting crop productivity globally, and innovative fertilization approaches are required to enhance nutrient use efficiency (NUE) and mitigate stress-induced yield losses. The authors synthesize ZnO and SiO₂ nanoparticles and apply them in a split-plot field experiment. Results indicate that nano Zn and Si significantly enhance biomass production, cob length, and grain yield, with nano Si outperforming other treatments, particularly under saline conditions.

The manuscript is well-structured, with a logical flow from introduction to conclusion. The study’s originality lies in its attempt to bridge the gap between greenhouse-based nano-fertilization research and real-world field applications under varying salinity conditions. The findings contribute to sustainable agricultural practices by offering an alternative strategy for enhancing nutrient uptake and crop resilience in saline-affected regions.

Strong Points

This study effectively investigates the efficacy of nano and conventional zinc and silicon fertilizers in maize cultivation under saline conditions, addressing an important issue in sustainable agriculture. The experimental design integrates nanotechnology with nutrient management strategies, offering valuable insights into enhancing crop resilience. The authors employ robust analytical techniques, including PCA and heat maps, to support their findings. Additionally, the comprehensive assessment of agronomic, biochemical, and physiological parameters strengthens the study’s impact, making it a relevant contribution to the field of plant nutrition under abiotic stress.

Weaker Aspects

In my opinion, the main handicapped points were the lack of a clearly stated hypothesis, an insufficient explanation of treatment rationale, and some gaps in methodological clarity.

One of the primary weaknesses of this study is the absence of a clearly stated hypothesis. The manuscript extensively discusses the effects of nano and conventional fertilizers on maize under saline stress, but it does not explicitly outline a testable hypothesis that links experimental treatments to expected outcomes. A well-structured hypothesis would help refine the study’s objectives and align the results with clear conclusions.

Furthermore, the conclusions drawn are somewhat broad and do not sufficiently tie back to a central hypothesis. While the discussion section effectively interprets the results, a stronger connection between the initial research aims and the findings is needed. It is recommended that the authors explicitly state their hypothesis in the introduction and reinforce it throughout the manuscript.

Below are my annotations with suggested improvements that the authors should carefully review to ensure accuracy.

Detailed Fixes 

  1. Page 1, Line 16 – The sentence "The increasing global population and worsening climate change have intensified challenges in sustainable agriculture..." is too broad. Suggest refining to directly relate to the study's scope.
  2. Page 1, Line 19 – "ZnO and SiO₂ nanoparticles were synthesized via co-precipitation and sol-gel methods, respectively..." It would be helpful to briefly explain why these methods were chosen.
  3. Page 1, Line 23 – The phrase "at recommended rates" is vague. Please specify exact application rates.
  4. Page 1, Line 25 – The expression "Nano Si exhibits the highest increase in biomass (110%) and nutrient use efficiency almost two times more than control..." should be reworded for clarity.
  5. Page 1, Line 28 – The PCA results are mentioned but not clearly linked to their biological significance.
  6. Page 2, Line 39 – The statement "Salinity, which affects approximately 20% of irrigated lands globally..." requires citation for verification.
  7. Page 2, Line 48 – The nutrient use efficiency (NUE) percentages need supporting references.
  8. Page 3, Line 65 – The study discusses various nanoparticles but does not specify their purity levels. Please provide details.
  9. Page 3, Line 75 – "Most studies have focused on greenhouse or laboratory experiments..." The authors should mention any prior field studies to better frame the novelty of their work.
  10. Page 3, Line 80 – The maize yield reductions mentioned in salinity-affected regions should include specific supporting references.
  11. Page 3, Line 86 – The research question and hypothesis should be explicitly stated to strengthen the study’s framework.
  12. Page 4, Line 92 – The abbreviation "conven. Zn" should be spelled out at least once before being abbreviated.
  13. Page 4, Line 106 – The sodium accumulation reduction mechanism should be explained further in relation to stress physiology.
  14. Page 5, Line 121 – The rationale for comparing nano Zn and nano Si instead of a broader range of elements should be clarified.
  15. Page 5, Line 140 – The phrase "Biological yield followed a similar trend..." is vague. Please specify exact percentages for comparison.
  16. Page 6, Line 171 – "Nano Zn and Si under non-saline conditions exhibited the highest positive influence..." The discussion should include an explanation of why these trends were observed.
  17. Page 7, Line 198 – The phrase "Sodium (Na⁺) and chloride (Cl⁻) ions accumulate in plant tissues, reducing water potential..." should include a reference to primary literature.
  18. Page 7, Line 210 – The phrase "Nano Zn and nano Si have shown particular effectiveness in alleviating salinity stress" should be supported by examples from past studies.
  19. Page 8, Line 230 – The discussion on Na⁺ exclusion via SOS1 antiporters needs a reference to support the claim.
  20. Page 8, Line 250 – "Nano Zn significantly increased Zn use efficiency in saline conditions by facilitating Zn uptake through rhizospheric pH modulation..." This is an important point but needs more details about the pH effect.
  21. Page 9, Line 264 – The conclusion states, "Nano Si strengthened cell walls, reduced oxidative stress, and improved water use efficiency..." Please provide supporting data from the results section.
  22. Page 10, Line 290 – The soil texture is mentioned as "sandy loam," but a more precise classification system (e.g., USDA soil taxonomy) should be used.
  23. Page 10, Line 304 – "Fertilizers were applied at recommended rates..." Please specify if these were based on prior field trials or established guidelines.
  24. Page 10, Line 329 – The formula for nutrient use efficiency should define variables clearly for improved readability.
  25. Page 12, Line 362 – The conclusion mentions "long-term soil interactions," but the study lacks a discussion on potential residual effects of nanoparticles on soil microbiology.

Comments on the Quality of English Language

The manuscript is generally well-written but contains minor grammatical errors and awkward phrasing that could affect readability. Below are some examples:

  1. Page 1, Line 28 – "These findings highlight the potential of nanotechnology-based fertilizers in mitigating salinity stress and enhancing crop productivity, offering a promising strategy for sustainable agriculture in salt-affected soils."

  2. Suggested fix: "These findings suggest that nanotechnology-based fertilizers can mitigate salinity stress and enhance crop productivity, providing a promising strategy for sustainable agriculture in salt-affected soils."
  3. Page 3, Line 60 – "Among these, nanoparticles of zinc (Zn) and silicon (Si) are particularly noteworthy for their ability to promote plant growth, improve nutrient uptake efficiency, and mitigate oxidative damage in crops exposed to abiotic stresses like salinity."

  4. Suggested fix: "Among these, Zn and Si nanoparticles are particularly effective in promoting plant growth, enhancing nutrient uptake efficiency, and reducing oxidative damage in crops under abiotic stress conditions."
  5. Page 5, Line 138 – "The results demonstrated significant improvements in yield parameters for all treatments compared to the control under both non-saline and saline conditions."

  6. Suggested fix: "Results showed significant yield improvements across all treatments compared to the control, under both saline and non-saline conditions."
  7. Page 8, Line 242 – "Additionally, our previous study on antioxidant enzyme response in maize under saline conditions confirmed that nano Zn and nano Si significantly boosted enzymatic activity, reducing oxidative stress markers such as MDA and H₂O₂."

  8. Suggested fix: "A prior study on maize antioxidant enzyme responses under saline conditions found that nano Zn and nano Si significantly enhanced enzymatic activity, reducing oxidative stress markers like MDA and H₂O₂."
  9. Page 10, Line 296 – "Soil texture was classified using the hydrometer method."

  10. Suggested fix: "Soil texture was determined using the hydrometer method."

Author Response

Thank you for your kind suggestion. Please see the attachment. All suggestion has been incorporated in final MS. 

Reviewer 2 Report

Comments and Suggestions for Authors

In the manuscript: “Efficacy of nano and conventional zinc and silicon fertilizers for nutrient use efficiency and yield benefits in maize under saline  field conditions“, authors

Abbas Shoukat, Uswah Mariyam, Britta Pitann, Muhammad Mubashar Zafar, Allah Nawaz, Waseem Hassan, Khalid M. Elhindi, Zulfiqar Ahmad Saqib, and Karl H. Mühling aims to provide actionable insights into the potential of nano technology-driven nutrient management to enhance crop performance and sustainability under real field conditions.

The paper is well written and is a worthy contribution that will be of interest to Plants readers. However, the paper has to be improved.

Abstract OK, just check the position of commas!

The abstract can serve as a stand-alone document that succinctly describes potential use and conclusions.

Keywords: OK

Introduction

L 41 delete comma before parenthesis

L 48 NUE in salt-stressed 48… write all words!

Ok, enough long and concise!

Materials and methods

L 298- 299 The detailed soil properties are provided in Table 1. or Table 3???

Table 3: Write formulas of the chemical in the right way!

OK, It is written on the way, that can be repeated! 

Results

L 92 (Tab 2). … Table 2 or at least a full stop. The same is true for Fig!

Do not use abbreviations in the Result section!

Legend is missing under the Figures and Tables 1 and 2!!!

Discussion

L193-213The first two paragraphs in the discussion do not fit into the discussion section because they are too general.

Conclusions:

Make it more concise.

Specific comments

Check the way you write references in the text; sometimes there is space between numbers and sometimes not! The article is very interesting, and it has many interesting results. It also has applicative value.

My comment: minor revision

Round 2

Reviewer 1 Report

Comments and Suggestions for Authors

Dear All,

I agree with the modifications made to the writing: “Efficacy of nano and conventional zinc and silicon fertilizers for nutrient use efficiency and yield benefits in maize under saline field conditions”. The authors have successfully addressed my comments and incorporated the suggested improvements effectively. I appreciate their thorough responses and thoughtful revisions. Therefore, I am happy to endorse this manuscript for publication.

Kind regards.

Author Response

Thank you for your valuable feedback. We have carefully addressed all the concerns raised and have made the necessary revisions as follows:

  1. Statistical Explanation (Section 4.9 & Figure Captions)

    • We have clarified the meaning of the error bars, specifying that they represent standard error (SE).
    • The letters above the bars in the figures are now explicitly explained as indicators of significant differences among treatments, determined using the Least Significant Difference (LSD) test after ANOVA.
    • We have mentioned the post hoc test (LSD test) used for mean comparisons after ANOVA in Section 4.9 and figure captions.
  2. Figure Revisions

    • Figures 1-4: We have included explanations for the statistical symbols (error bars and lettering) in the captions to ensure clarity.
    • Figure 5:
      • Abbreviations for agronomic parameters have been defined in the figure caption, similar to Figure 6.
      • The color scale (cos2) has been explained, specifying that it represents the quality of representation of variables in the PCA space, with higher values indicating stronger contributions and lower values indicating weaker contributions to the principal components.

These revisions ensure that all figures can be understood independently of the main text and that statistical methodologies are clearly documented. We appreciate the reviewer's insights, which have helped improve the clarity of our manuscript.